# Pattern Selection for Optimal Classical Planning with Saturated Cost Partitioning

**Jendrik Seipp**
University of Basel
Basel, Switzerland
jendrik.seipp@unibas.ch

## Abstract

Pattern databases are the foundation of some of the strongest admissible heuristics for optimal classical planning. Experiments showed that the most informative way of combining information from multiple pattern databases is to use saturated cost partitioning. Previous work selected patterns and computed saturated cost partitionings over the resulting pattern database heuristics in two separate steps. We introduce a new method that uses saturated cost partitioning to select patterns and show that it outperforms all existing pattern selection algorithms.

## 1    Introduction

$A^*$ search (Hart, Nilsson, and Raphael 1968) with an admissible heuristic (Pearl 1984) is one of the most successful methods for solving classical planning tasks optimally. An important building block of some of the strongest admissible heuristics are pattern database (PDB) heuristics. A PDB heuristic precomputes all goal distances in a simplified state space obtained by projecting the task to a subset of state variables, the *pattern*, and uses these distances as lower bounds on the true goal distances. PDB heuristics were originally introduced for solving the 15-puzzle (Culberson and Schaeffer 1996) and have later been generalized to many other combinatorial search tasks (e.g., Korf 1997; Felner, Korf, and Hanan 2004) and to the setting of domain-independent planning (Edelkamp 2001).

Using a single PDB heuristic of reasonable size is usually not enough to cover sufficiently many aspects of challenging planning tasks. It is therefore often beneficial to compute multiple PDB heuristics and to combine their estimates admissibly (Holte et al. 2006). The simplest approach for this is to choose the PDB with the highest estimate in each state. Instead of this maximization scheme, we would like to sum estimates, but this renders the resulting heuristic inadmissible in general. However, if two PDBs are affected by disjoint sets of operators, they are *independent* and we can admissibly add their estimates (Korf and Felner 2002; Felner, Korf, and Hanan 2004). Haslum et al. (2007) later generalized this idea by introducing the *canonical heuristic* for PDBs, which computes all maximal subsets of pairwise independent PDBs and then uses the maximum over the sums of independent PDBs as the heuristic value.

*Cost partitioning* (Katz and Domshlak 2008; Yang et al. 2008) is a generalization of the independence-based methods above. It makes the sum of heuristic estimates admissible by distributing the costs of each operator among the heuristics. The literature contains many different cost partitioning algorithms such as zero-one cost partitioning (Edelkamp 2002; Haslum et al. 2007), uniform cost partitioning (Katz and Domshlak 2008), optimal cost partitioning (Katz and Domshlak 2008; Karpas and Domshlak 2009; Katz and Domshlak 2010; Pommerening et al. 2015), post-hoc optimization (Pommerening, Röger, and Helmert 2013) and delta cost partitioning (Fan, Müller, and Holte 2017).

In previous work (Seipp, Keller, and Helmert 2017a), we showed experimentally for the benchmark tasks from previous International Planning Competitions (IPC) that *saturated cost partitioning* (SCP) (Seipp and Helmert 2014; 2018) is the cost partitioning algorithm of choice for PDB heuristics. Saturated cost partitioning considers an ordered sequence of heuristics. Iteratively, it gives each heuristic the minimum amount of costs that the heuristic needs to justify all its estimates and then uses the remaining costs for subsequent heuristics until all heuristics have been served this way.

Before we can compute a saturated cost partitioning over pattern database heuristics, we need to select a collection of patterns. The first domain-independent automated pattern selection algorithm is due to Edelkamp (2001). It partitions the state variables into patterns via best-fit bin packing. Edelkamp (2006) later used a genetic algorithm to search for a pattern collection that maximizes the average heuristic value of a zero-one cost partitioning over the PDB heuristics.

Haslum et al. (2007) proposed an algorithm that performs a hill-climbing search in the space of pattern collections (HC). HC evaluates a collection $C$ by estimating the search effort of the canonical heuristic over $C$ based on a model of IDA$^*$ runtime (Korf, Reid, and Edelkamp 2001).

Franco et al. (2017) presented the Complementary PDBs Creation (CPC) method, that combines bin packing and genetic algorithms to create a pattern collection minimizing the estimated search effort of an A$^*$ search (Lelis, Stern, and Sturtevant 2014).

Rovner, Sievers, and Helmert (2019) repeatedly compute patterns using counterexample-guided abstraction refinement (CEGAR): starting from a random goal variable,

their CEGAR algorithm iteratively finds solutions in the corresponding projection and executes them in the original state space. Whenever a solution cannot be executed due to a violated precondition, it adds the missing precondition variable to the pattern.

Finally, Pommerening, Röger, and Helmert (2013) systematically generate all *interesting* patterns up to a given size $X$ (SYS-X). Experiments showed that cost-partitioned heuristics over SYS-2 and SYS-3 yield accurate estimates (Pommerening, Röger, and Helmert 2013; Seipp, Keller, and Helmert 2017a), but using all interesting patterns of larger sizes is usually infeasible.

We introduce SYS-SCP, a new pattern selection algorithm based on saturated cost partitioning that potentially considers all interesting patterns, but only selects useful ones. SYS-SCP builds multiple pattern sequences that together form the resulting pattern collection. For each sequence $\sigma$, it considers the interesting patterns in increasing order by size and adds a pattern $P$ to $\sigma$ if $P$ is not part of an earlier sequence and the saturated cost partitioning heuristic over $\sigma$ plus $P$ is more informative than the one over $\sigma$ alone.

## 2   Background

We consider optimal classical planning tasks in a $SAS^+$-like notation (Bäckström and Nebel 1995) and represent a planning task $\Pi$ as a tuple $\langle \mathcal{V}, \mathcal{O}, s_0, s_\star \rangle$. Each variable $v$ in the finite set of variables $\mathcal{V}$ has a finite domain $dom(v)$. A *partial state* $s$ is defined over a subset of variables $vars(s) \subseteq \mathcal{V}$ and maps each $v \in vars(s)$ to a value in $dom(v)$, written as $s[v]$. We call the pair $\langle v, s[v] \rangle$ an *atom* and interchangeably treat partial states as mappings from variables to values or as sets of atoms. If $vars(s) = \mathcal{V}$, we call $s$ a *state*. We write $S(\Pi)$ for the set of all states in $\Pi$.

Each operator $o$ in the finite set of operators $\mathcal{O}$ has a *precondition pre(o)* and an *effect eff(o)*, both of which are partial states, and a cost $cost(o) \in \mathbb{R}_0^+$. An operator $o$ is applicable in a state $s$ if $pre(o) \subseteq s$. Applying $o$ in $s$ leads into state $s' = s[\![o]\!]$ with $s'[v] = eff(o)[v]$ for all $v \in vars(eff(o))$ and $s'[v] = s[v]$ for all variables $v \in \mathcal{V} \setminus vars(eff(o))$. The state $s_0$ is called the *initial state* and $s_\star$ is a partial state, the *goal*.

Transition systems assign semantics to planning tasks.

**Definition 1** (Transition Systems). *A transition system $\mathcal{T}$ is a labeled digraph defined by a finite set of* states $S(\mathcal{T})$*, a finite set of* labels $L(\mathcal{T})$*, a set $T(\mathcal{T})$ of labeled* transitions $s \xrightarrow{\ell} s'$ *with $s, s' \in S(\mathcal{T})$ and $\ell \in L(\mathcal{T})$, an* initial state $s_0(\mathcal{T})$*, and a set $S_\star(\mathcal{T})$ of goal states.*

A planning task $\Pi = \langle \mathcal{V}, \mathcal{O}, s_0, s_\star \rangle$ induces a transition system $\mathcal{T}$ with states $S(\Pi)$, labels $\mathcal{O}$, transitions $\{s \xrightarrow{o} s[\![o]\!] \mid s \in S(\Pi), o \in \mathcal{O}, pre(o) \subseteq s\}$, initial state $s_0$ and goal states $\{s \in S(\Pi) \mid s_\star \subseteq s\}$.

Separating transition systems from *cost functions* allows us to evaluate the same transition system under different cost functions, which is important for cost partitioning.

**Definition 2** (Cost Functions). *A* cost function *for transition system $\mathcal{T}$ is a function cost $: L(\mathcal{T}) \to \mathbb{R} \cup \{-\infty, \infty\}$. It is*

finite *if $-\infty < cost(\ell) < \infty$ for all labels $\ell$. It is* non-negative *if $cost(\ell) \geq 0$ for all labels $\ell$. We write $\mathcal{C}(\mathcal{T})$ for the set of all cost functions for $\mathcal{T}$.*

Note that we assume that the cost function of the planning task is non-negative and finite, but as in previous work we allow negative (Pommerening et al. 2015) and infinite costs (Seipp and Helmert 2019) in cost partitionings. The generalization to infinite costs is necessary to cleanly state some of our definitions.

**Definition 3** (Weighted Transition Systems). *A* weighted transition system *is a pair $\langle \mathcal{T}, cost \rangle$ where $\mathcal{T}$ is a transition system and $cost \in \mathcal{C}(\mathcal{T})$ is a cost function for $\mathcal{T}$.*

The *cost* of a *path* $\pi = \langle s^0 \xrightarrow{\ell_1} s^1, \ldots, s^{n-1} \xrightarrow{\ell_n} s^n \rangle$ in a weighted transition system $\langle \mathcal{T}, cost \rangle$ is defined as $cost(\pi) = \sum_{i=1}^n cost(\ell_i)$. It is $\infty$ if the sum contains both $+\infty$ and $-\infty$. If $s^n$ is a goal state, $\pi$ is called a *goal path* for $s^0$.

**Definition 4** (Goal Distances and Optimal Paths). *The goal distance of a state $s \in S(\mathcal{T})$ in a weighted transition system $\langle \mathcal{T}, cost \rangle$ is defined as $\inf_{\pi \in \Pi_\star(\mathcal{T}, s)} cost(\pi)$, where $\Pi_\star(\mathcal{T}, s)$ is the set of goal paths from $s$ in $\mathcal{T}$. (The infimum of the empty set is $\infty$.) We write $h_\mathcal{T}^*(cost, s)$ for the goal distance of $s$. If $h_\mathcal{T}^*(cost, s) = \infty$, we call $s$* unsolvable. *A goal path $\pi$ from $s$ is* optimal *if $cost(\pi) = h_\mathcal{T}^*(cost, s)$.*

Optimal classical planning is the problem of finding an optimal goal path from $s_0$ or showing that $s_0$ is unsolvable.

We use *heuristics* to estimate goal distances (Pearl 1984).

**Definition 5** (Heuristics). *A heuristic for a transition system $\mathcal{T}$ is a function $h : \mathcal{C}(\mathcal{T}) \times S(\mathcal{T}) \to \mathbb{R} \cup \{-\infty, \infty\}$. Heuristic $h$ is* admissible *if $h(cost, s) \leq h_\mathcal{T}^*(cost, s)$ for all $cost \in \mathcal{C}(\mathcal{T})$ and all $s \in S(\mathcal{T})$.*

Cost partitioning makes adding heuristics admissible by distributing the costs of each operator among the heuristics.

**Definition 6** (Cost Partitioning). *Let $\mathcal{T}$ be a transition system. A* cost partitioning *for a cost function $cost \in \mathcal{C}(\mathcal{T})$ is a tuple $\langle cost_1, \ldots, cost_n \rangle \in \mathcal{C}(\mathcal{T})^n$ whose sum is bounded by cost: $\sum_{i=1}^n cost_i(\ell) \leq cost(\ell)$ for all $\ell \in L(\mathcal{T})$. A cost partitioning $\langle cost_1, \ldots, cost_n \rangle \in \mathcal{C}(\mathcal{T})^n$ over the heuristics $\langle h_1, \ldots, h_n \rangle$ for $\mathcal{T}$ induces the* cost-partitioned heuristic $h(cost, s) = \sum_{i=1}^n h_i(cost_i, s)$. *If the sum contains $+\infty$ and $-\infty$, it evaluates to the leftmost infinite value.*

One of the cost partitioning algorithms from the literature is *saturated cost partitioning* (Seipp and Helmert 2018). It is based on the insight that we can often reduce the amount of costs given to a heuristic without changing any heuristic estimates. *Saturated cost functions* formalize this idea.

**Definition 7** (Saturated Cost Function). *Consider a transition system $\mathcal{T}$, a heuristic $h$ for $\mathcal{T}$ and a cost function $cost \in \mathcal{C}(\mathcal{T})$. A cost function $scf \in \mathcal{C}(\mathcal{T})$ is* saturated *for $h$ and cost if*

1. *$scf(\ell) \leq cost(\ell)$ for all labels $\ell \in L(\mathcal{T})$ and*
2. *$h(scf, s) = h(cost, s)$ for all states $s \in S(\mathcal{T})$.*

*A saturated cost function $scf$ is* minimal *if there is no other saturated cost function $scf'$ for $h$ and cost with $scf(\ell) \leq scf'(\ell)$ for all labels $\ell \in L(\mathcal{T})$.*

Whether we can efficiently compute a minimal saturated cost function depends on the type of heuristic. In earlier work (Seipp and Helmert 2018), we showed that this is possible for explicitly-represented abstraction heuristics (Helmert, Haslum, and Hoffmann 2007), which include PDB heuristics.

**Definition 8** (Minimum Saturated Cost Function for Abstraction Heuristics). *Let $\langle \mathcal{T}, cost \rangle$ be a weighted transition system and $h$ an abstraction heuristic for $\mathcal{T}$ with abstract transition system $\mathcal{T}'$. The* minimum saturated cost function *mscf for $h$ and cost is*

$$mscf(\ell) = \sup_{a \xrightarrow{\ell} b \in T(\mathcal{T}')} (h^*_{\mathcal{T}'}(cost, a) - h^*_{\mathcal{T}'}(cost, b))$$

*for all $\ell \in L(\mathcal{T})$, where $x - y = -\infty$ iff $x = -\infty$ or $y = \infty$.*

Given a sequence of abstraction heuristics, the saturated cost partitioning algorithm iteratively assigns to each heuristic only the costs that the heuristic needs to preserve its estimates and uses the remaining costs for subsequent heuristics.

**Definition 9** (Saturated Cost Partitioning). *Consider a transition system $\mathcal{T}$ and a sequence of abstraction heuristics $\mathcal{H} = \langle h_1, \ldots, h_n \rangle$ for $\mathcal{T}$. For all $1 \leq i \leq n$, $saturate_i : \mathcal{C}(\mathcal{T}) \to \mathcal{C}(\mathcal{T})$ receives a cost function rem and returns the minimum saturated cost function for $h_i$ and rem. The* saturated cost partitioning $\langle cost_1, \ldots, cost_n \rangle$ *of a function $cost \in \mathcal{C}(\mathcal{T})$ over $\mathcal{H}$ is defined as:*

$$rem_0 = cost$$
$$cost_i = saturate_i(rem_{i-1}) \qquad \text{for all } 1 \leq i \leq n$$
$$rem_i = rem_{i-1} - cost_i \qquad \text{for all } 1 \leq i \leq n,$$

*where the auxiliary cost functions $rem_i$ represent the remaining costs after processing the first $i$ heuristics in $\mathcal{H}$.*

We write $h^{\mathrm{SCP}}_{\mathcal{H}}$ for the saturated cost partitioning heuristic over the sequence of heuristics $\mathcal{H}$. In this work, we compute saturated cost partitionings over pattern database heuristics.

A *pattern* for task $\Pi$ with variables $\mathcal{V}$ is a subset $P \subseteq \mathcal{V}$. By syntactically removing all variables from $\Pi$ that are not in $P$, we obtain the *projected* task $\Pi|_P$ inducing the abstract transition system $\mathcal{T}_P$. The PDB heuristic $h^P$ for a pattern $P$ is defined as $h^P(cost, s) = h^*_{\mathcal{T}_P}(cost, s|_P)$, where $s|_P$ is the abstract state that $s$ is projected to in $\Pi|_P$. For the pattern sequence $\langle P_1, \ldots, P_n \rangle$ we define $h^{\mathrm{SCP}}_{\langle P_1, \ldots, P_n \rangle} = h^{\mathrm{SCP}}_{\langle h^{P_1}, \ldots, h^{P_n} \rangle}$.

One of the simplest pattern selection algorithms is to generate all patterns up to a given size $X$ (Felner, Korf, and Hanan 2004) and we call this approach SYS-NAIVE-X. It is easy to see that for tasks with $n$ variables, SYS-NAIVE-X generates $\sum_{i=1}^{X} \binom{n}{i}$ patterns. Usually, many of these patterns do not add much information to a cost-partitioned heuristic over the patterns. Unfortunately, there is no efficiently computable test that allows us to discard such uninformative patterns. Even patterns without any goal variables can increase heuristic estimates in a cost partitioning (Pommerening 2017).

However, in the setting where only non-negative cost functions are allowed in cost partitionings, there are efficiently computable criteria for deciding whether a pattern

---

**Algorithm 1** SYS-SCP: Given a planning task with states $S(\mathcal{T})$, cost function *cost* and interesting patterns SYS, select a subset $C \subseteq$ SYS.

1: **function** SYS-SCP($\Pi$)
2:     $C \leftarrow \emptyset$
3:     **repeat** for at most $T_x$ seconds
4:         $\sigma \leftarrow \langle \rangle$
5:         **for** $P \in$ ORDER(SYS) **and** at most $T_y$ seconds **do**
6:             **if** $P \notin C$ **and** PATTERNUSEFUL($\sigma, P$) **then**
7:                 $\sigma \leftarrow \sigma \oplus P$
8:                 $C \leftarrow C \cup \{P\}$
9:     **until** $\sigma = \langle \rangle$
10:     **return** $C$

11: **function** PATTERNUSEFUL($\sigma, P$)
12:     **return** $\exists s \in S(\mathcal{T}) :$
            $h^{\mathrm{SCP}}_{\sigma}(cost, s) < h^{\mathrm{SCP}}_{\sigma \oplus P}(cost, s) < \infty$

---

is *interesting*, i.e., whether it cannot be replaced by a set of smaller patterns that together yield the same heuristic estimates (Pommerening, Röger, and Helmert 2013).

The criteria are based on the *causal graph* $CG(\Pi)$ of a task $\Pi$ (Helmert 2004). $CG(\Pi)$ is a directed graph with a node for each variable in $\Pi$. If there is an operator with a precondition on $u$ and an effect on $v \neq u$, $CG(\Pi)$ contains a *precondition* arc from $u$ to $v$. If an operator affects both $u$ and $v$, $CG(\Pi)$ contains *co-effect* arcs from $u$ to $v$ and from $v$ to $u$.

**Definition 10** (Interesting Patterns). *A pattern $P$ is interesting if*

1. *$CG(\Pi|_P)$ is weakly connected, and*
2. *$CG(\Pi|_P)$ contains a directed path via precondition arcs from each node to some goal variable node.*

The systematic pattern generation method SYS-X generates all interesting patterns up to size $X$. We let SYS denote the set of all interesting patterns for a given task. On IPC benchmark tasks, SYS-X often generates much fewer patterns than SYS-NAIVE-X for the same size limit $X$. Still, it is usually infeasible to compute all SYS-X patterns and the corresponding projections for $X > 3$ within reasonable amounts of time and memory. Also, we hypothesize that even when considering only interesting patterns, usually only a small percentage of the systematic patterns up to size 3 contribute much information to the resulting heuristic.

For these two reasons we propose a new pattern selection algorithm that potentially considers all interesting patterns, but only selects the ones that it deems useful.

## 3 Sys-SCP Pattern Selection Algorithm

Our new pattern selection algorithm repeatedly creates a new empty pattern sequence $\sigma$ and only appends those interesting patterns to $\sigma$ that increase any finite heuristic values of a saturated cost partitioning heuristic computed over $\sigma$.

Algorithm 1 shows pseudo-code for the procedure, which we call SYS-SCP. It starts with an empty pattern collection $C$. In each iteration of the outer loop, SYS-SCP creates a

new empty pattern sequence $\sigma$ and then loops over the interesting patterns $P \in \text{SYS}$ in the order chosen by ORDER (see Section 3.2) for at most $T_y$ seconds. SYS-SCP appends a pattern $P$ to $\sigma$ and includes it in $C$ if there is a state $s$ for which the saturated cost partitioning over $\sigma$ extended by $P$ has a higher finite heuristic value than the one over $\sigma$ alone. Once an iteration selects no new patterns or SYS-SCP hits the time limit $T_x$, the algorithm stops and returns $C$.

We impose a time limit $T_x$ on the outer loop of the algorithm since the number of interesting patterns is exponential in the number of variables and therefore SYS-SCP usually cannot evaluate them all in a reasonable amount of time. By imposing a time limit $T_y$ on the inner loop, we allow SYS-SCP to periodically start over with a new empty pattern sequence.

The most important component of the SYS-SCP algorithm is the PATTERNUSEFUL function that decides whether to select a pattern $P$. The function enumerates all states $s \in S(\Pi)$, which is obviously infeasible for all but the smallest tasks $\Pi$. Fortunately, we can efficiently compute an equivalent test in the projection to $P$.

**Lemma 1.** *Consider a planning task $\Pi$ with non-negative cost function cost and induced transition system $\mathcal{T}$. Let $s \in S(\mathcal{T})$ be a state, $P$ be a pattern for $\Pi$ and $\sigma$ be a (possibly empty) sequence of patterns $\langle P_1, \dots, P_n \rangle$ for $\Pi$. Finally, let rem be the remaining cost function after computing $h_\sigma^{SCP}$ for cost.*

$$h_\sigma^{SCP}(cost, s) < h_{\sigma \oplus P}^{SCP}(cost, s) < \infty$$
$$\Leftrightarrow 0 < h_{\mathcal{T}_P}^*(rem, s|_P) < \infty$$

*Proof.* $h_\sigma^{\text{SCP}}(cost, s) < h_{\sigma \oplus P}^{\text{SCP}}(cost, s) < \infty$

$\overset{(1)}{\Leftrightarrow} h_{\langle P_1, \dots, P_n \rangle}^{\text{SCP}}(cost, s) < h_{\langle P_1, \dots, P_n, P \rangle}^{\text{SCP}}(cost, s) < \infty$

$\overset{(2)}{\Leftrightarrow} \sum_{i=1}^n h^{P_i}(cost_i, s) < \sum_{i=1}^n h^{P_i}(cost_i, s) + h^P(rem, s) < \infty$

$\overset{(3)}{\Leftrightarrow} 0 < h^P(rem, s) < \infty \overset{(4)}{\Leftrightarrow} 0 < h_{\mathcal{T}_P}^*(rem, s|_P) < \infty$

Step 1 substitutes $\langle P_1, \dots, P_n \rangle$ for $\sigma$ and Step 2 uses the definition of saturated cost partitioning heuristics. For Step 3 we need to show that $x = \sum_{i=1}^n h^{P_i}(cost_i, s)$ is finite.

The inequality states $x < \infty$. We now show $x \geq 0$, which implies $x > -\infty$. Using requirement 1 for saturated cost functions from Definition 7 and the fact that $rem_0 = cost$ is non-negative, it is easy to see that all remaining cost functions are non-negative. Consequently, $h^{P_i}(cost_i, s) = h^{P_i}(rem_{i-1}, s) \geq 0$ for all $s \in S(\mathcal{T})$, which uses requirement 2 from Definition 7 and the fact that goal distances are non-negative in transition systems with non-negative weights.

Step 4 uses the definition of PDB heuristics. $\square$

**Theorem 1** (Computing PATTERNUSEFUL on Projections). *Consider a planning task $\Pi$ with non-negative cost function cost and induced transition system $\mathcal{T}$. Let $P$ be a single pattern and $\sigma$ be a (possibly empty) sequence of patterns. Finally, let rem be the remaining cost function after computing*

$h_\sigma^{SCP}$ *for cost.*

$$\exists s \in S(\mathcal{T}) : h_\sigma^{SCP}(cost, s) < h_{\sigma \oplus P}^{SCP}(cost, s) < \infty$$
$$\Leftrightarrow \exists s' \in S(\mathcal{T}_P) : 0 < h_{\mathcal{T}_P}^*(rem, s') < \infty$$

*Proof.* Follows directly from Lemma 1 and the fact that projections are induced abstractions: for each abstract state $s'$ in an induced abstraction there is at least one concrete state $s$ which is projected to $s'$. $\square$

We use Theorem 1 in our SYS-SCP implementation by keeping track of the cost function *rem*, i.e., the costs that remain after computing $h_\sigma^{\text{SCP}}$. We select a pattern $P$ if there are any goal distances $d$ with $0 < d < \infty$ in $\mathcal{T}_P$ under *rem*.

Theorem 1 also removes the need to compute $h_{\sigma \oplus P}^{\text{SCP}}$ from scratch for every pattern $P$. This is important since we want to decide whether or not to add $P$ quickly and this operation should not become slower when $\sigma$ contains more patterns.

### 3.1 Dead Ends

To obtain high finite heuristic values for solvable states it is important to choose good cost partitionings. In contrast, cost functions are irrelevant for detecting unsolvable states. This is the underlying reason why Lemma 1 only holds for finite values and therefore why SYS-SCP ignores unsolvable states.

However, we can still use the information about unsolvable states contained in projections. It is easy to see that each abstract state in a projection corresponds to a partial state in the original task. If an abstract state is unsolvable in a projection, we call the corresponding partial state a *dead end*. Since projections preserve all paths, any state in the original task subsuming a dead end is unsolvable. We can extract all dead ends from the projections that SYS-SCP evaluates and use this information to prune unsolvable states during the $A^*$ search (Pommerening and Seipp 2016).

### 3.2 Ordering Patterns

We showed in earlier work that the order in which saturated cost partitioning considers the component heuristics has a strong influence on the quality of the resulting heuristic (Seipp, Keller, and Helmert 2017b). Choosing a good order is even more important for SYS-SCP, since it usually only sees a subset of interesting patterns within the allotted time. To ensure that this subset of interesting patterns covers different aspects of the planning task, we let the ORDER function generate the interesting patterns in increasing order by size.

This leaves the question how to sort patterns of the same size. We propose four methods for making this decision. The first one (*random*) simply orders patterns of the same size randomly. The remaining three assign a key to each pattern, allowing us to sort by key in increasing or decreasing order.

**Causal Graph.** The first ordering method is based on the insight that it is often more important to have accurate heuristic estimates near the goal states rather than elsewhere in the state space (e.g., Holte et al. 2006; Torralba,

Linares López, and Borrajo 2018). We therefore want to focus on patterns containing goal variables or variables that are closely connected to goal variables. To quantify "goal-connectedness" we use an approximate topological ordering $\prec$ of the causal graph $CG(\Pi)$. We let the function $cg : \mathcal{V} \to \mathbb{N}_0^+$ assign each variable $v \in \mathcal{V}$ to its index in $\prec$. For a given pattern $P$, the $cg$ ordering method returns the key $\langle cg(v_1), \ldots, cg(v_n) \rangle$, where $v_i \in P$ and $cg(v_i) < cg(v_j)$ for all $1 \leq i < j \leq n$. Since the keys are unique, they define a total order. Sorting the patterns by $cg$ in decreasing order (*cg-down*), yields the desired order which starts with "goal-connected" patterns.

**States in Projection.** Given a pattern $P$, the ordering method *states* returns the key $|S(\Pi|_P)|$, i.e., the number of states in the projection to $P$. We use *cg-down* to break ties.

**Active Operators.** Given a pattern $P$, the *ops* ordering method returns the number of operators that affect a variable in $P$. We break ties with *cg-down*.

## 4 Experiments

We implemented the SYS-SCP pattern selection algorithm in the Fast Downward planning system (Helmert 2006) and conducted experiments with the Downward Lab toolkit (Seipp et al. 2017) on Intel Xeon Silver 4114 processors. Our benchmark set consists of all 1827 tasks without conditional effects from the optimization tracks of the 1998–2018 IPCs. The tasks belong to 48 different domains. We limit time by 30 minutes and memory by 3.5 GiB. All benchmarks[1], code[2] and experimental data[3] have been published online.

To fairly compare the quality of different pattern collections, we use the same cost partitioning algorithm for all collections. Saturated cost partitioning is the obvious choice for the evaluation since experiments showed that it is preferable to all other cost partitioning algorithms for HC, SYS-2 and CPC patterns in almost all evaluated benchmark domains (Seipp, Keller, and Helmert 2017a; Rovner, Sievers, and Helmert 2019).

**Diverse Saturated Cost Partitioning Heuristics.** For a given pattern collection $C$, we compute diverse saturated cost partitioning heuristics using the diversification procedure by Seipp, Keller, and Helmert (2017b): we start with an empty family of saturated cost partitioning heuristics $\mathcal{F}$ and a set $\hat{S}$ of 1000 sample states obtained with random walks (Haslum et al. 2007). Then we iteratively sample a new state $s$ and compute a *greedy* order $\omega$ of $C$ that works well for $s$ (Seipp 2017). If $h_\omega^{\text{SCP}}$ has a higher heuristic estimate for any state $s' \in \hat{S}$ than all heuristics in $\mathcal{F}$, we add $h_\omega^{\text{SCP}}$ to $\mathcal{F}$. We stop this diversification procedure after 200 seconds and then perform an A* search using the maximum over the heuristics in $\mathcal{F}$.

---

[1]Benchmarks: https://doi.org/10.5281/zenodo.2616479
[2]Code: https://doi.org/10.5281/zenodo.3233330
[3]Experimental data: https://doi.org/10.5281/zenodo.3233326

| Coverage | 10s | 100s | 1000s | $\infty$ |
|---|---|---|---|---|
| 1s | **1137** | 1132 | 1055 | 716 |
| 10s | 1077 | **1168** | 1142 | 337 |
| 100s | 1077 | 1082 | **1154** | 284 |
| $\infty$ | 1077 | **1082** | 989 | 227 |

Table 1: Number of tasks solved by SYS-SCP using different time limits $T_x$ and $T_y$ for the outer loop ($x$ axis) and inner loop ($y$ axis).

| | cg-up | states-up | random | ops-down | states-down | ops-up | cg-down | Coverage |
|---|---|---|---|---|---|---|---|---|
| cg-up | – | 5 | 6 | 5 | 4 | 3 | 3 | 1140.0 |
| states-up | **6** | – | 6 | **8** | 5 | 2 | 2 | 1153.0 |
| random | **10** | **10** | – | 8 | 7 | 6 | 3 | 1148.2 |
| ops-down | **7** | **8** | **9** | – | 4 | 7 | 3 | 1141.0 |
| states-down | **9** | **8** | **9** | **7** | – | 4 | 2 | 1152.0 |
| ops-up | **11** | **12** | **12** | **11** | **11** | – | 6 | 1166.0 |
| cg-down | **12** | **10** | **12** | **10** | **9** | **6** | – | **1168.0** |

Table 2: Per-domain coverage comparison of different orders for patterns of the same size. The entry in row $r$ and column $c$ shows the number of domains in which order $r$ solves more tasks than order $c$. For each order pair we highlight the maximum of the entries $(r, c)$ and $(c, r)$ in bold. Right: Total number of solved tasks. The results for *random* are averaged over 10 runs (standard deviation: 3.36).

Before we compare SYS-SCP to other pattern selection algorithms, we evaluate the effects of changing its parameters in four ablation studies. We use at most 2M states per PDB and 20M states in the PDB collection for all SYS-SCP runs.

### 4.1 Time Limits

Table 1 shows that a time limit for the outer loop is more important than one for the inner loop, but for maximum coverage we need both limits. The combination that solves the highest number of tasks is 10s for the inner and 100s for the outer loop. We use these values in all other experiments.

### 4.2 Dead Ends

All configurations from Table 1 store the dead ends from all projections evaluated by SYS-SCP and use them to prune unsolvable states during the A* search. For the best configuration from Table 1, coverage decreases from 1168 to 1153 tasks if we ignore the dead ends. Therefore, we use dead ends for pruning unsolvable states in all other experiments.

### 4.3 Pattern Orders

Table 2 compares the different methods for ordering patterns of the same size. For all of *states*, *ops* and *cg*, at least one or-

| Max pattern size | 1 | 2 | 3 | 4 | 5 |
|---|---|---|---|---|---|
| SYS-NAIVE | 840 | **937** | 914 | 752 | 571 |
| SYS-NAIVE-LIM | 840 | 968 | **1004** | 912 | 878 |
| SYS | 840 | 986 | **1057** | 922 | 731 |
| SYS-LIM | 840 | 985 | **1088** | 1050 | 1035 |

Table 3: Number of solved tasks for naive (SYS-NAIVE) and interesting patterns (SYS). We evaluate both versions without and with time and memory limits and using different maximum pattern sizes.

| | HC | SYS-3-LIM | CPC | CEGAR | SYS-SCP | Coverage |
|---|---|---|---|---|---|---|
| HC | – | 8 | 10 | 8 | 3 | 966 |
| SYS-3-LIM | 19 | – | 14 | 10 | 2 | 1088 |
| CPC | 20 | 15 | – | 12 | 3 | 1055 |
| CEGAR | 22 | 14 | 16 | – | 3 | 1098 |
| SYS-SCP | 28 | 23 | 21 | 21 | – | **1168** |

Table 4: Per-domain coverage comparison of pattern selection algorithms. For an explanation of the data see the caption of Table 2.

dering direction (*up* or *down*) is preferable to using random orders. The *ops-up* method is preferable to *ops-down* for 11 domains, but there are also 7 domains where the opposite is the case. The relation between *states-down* and *states-up* is similar. The only ordering method where one direction is clearly preferable to the other is *cg*: *cg-down* solves more tasks than *cg-up* in 12 domains, while the opposite is the case in only 3 domains. Since *cg-down* also has the highest overall coverage, we use it in all other experiments.

### 4.4 Using Pattern Sequences for Diversification

Instead of discarding the computed pattern sequences when SYS-SCP finishes, we can turn each pattern sequence $\sigma$ into a full pattern order by randomly appending all SYS-SCP patterns missing from $\sigma$ to $\sigma$ and pass the resulting order to the diversification procedure.

Feeding the diversification exclusively with such orders leads to solving 1130 tasks, while using only greedy orders for sample states (Seipp 2017) solves 1156 tasks. We obtain the best results by diversifying both types of orders, solving 1168 tasks, and we use this variant in all other experiments.

### 4.5 Systematic Patterns With Limits

In the next experiment, we evaluate the obvious baseline for SYS-SCP: selecting all (interesting) patterns up to a fixed size. Table 3 holds coverage results of SYS-NAIVE-X and SYS-X for $1 \leq X \leq 5$. We also include variants (*-LIM) that use at most 100 seconds, no more than 2M states in each projection and at most 20M states per collection. For the *-LIM variants, we sort the patterns in the *cg-down* order.

The results show that interesting patterns are always preferable to naive patterns, both with and without limits, which is why we only consider interesting patterns in SYS-SCP. Imposing limits is not important for SYS-1 and SYS-2, but leads to solving many more tasks for $X \geq 3$. Overall, SYS-3-LIM has the highest total coverage (1088 tasks).

### 4.6 Comparison of Pattern Selection Algorithms

In Table 4 we compare SYS-SCP to the strongest pattern selection algorithms from the literature: HC, SYS-3-LIM, CPC and CEGAR. (See Table 6 for per-domain coverage results.) We run each algorithm with its preferred parameter values, which implies using at most 900s for HC and CPC and 100s for the other algorithms.

HC is outperformed by all other algorithms. Interestingly, already the simple SYS-3-LIM approach is competitive with

CPC and CEGAR. However, we obtain the best results with SYS-SCP. It is preferable to all other pattern selection algorithms in per-domain comparisons: no algorithm has higher coverage than SYS-SCP in more than three domains, while SYS-SCP solves more tasks than each of the other algorithms in at least 21 domains. SYS-SCP also has the highest total coverage of 1168 tasks, solving 70 more tasks than the strongest contender. This is a considerable improvement in the setting of optimal classical planning, where task difficulty tends to scale exponentially.

### 4.7 Comparison to IPC Planners

In our final experiment, we evaluate whether Scorpion (Seipp 2018), one of the strongest optimal planners in IPC 2018, benefits from using SYS-SCP patterns. Scorpion computes diverse saturated cost partitioning heuristics over HC and SYS-2 PDB heuristics and Cartesian abstraction heuristics (CART) (Seipp and Helmert 2018). We abbreviate this combination with COMB=HC+SYS-2+CART. In Table 5 we compare the original Scorpion planner, three Scorpion variants that use different sets of heuristics and the top three optimal planners from IPC 2018, Delfi 1 (Sievers et al. 2019), Complementary 1 (Franco et al. 2018) and Complementary 2 (Franco et al. 2017). (Table 6 holds per-domain coverage results.) In contrast to the configurations we evaluated above, all planners in Table 5 prune irrelevant operators in a preprocessing step (Alcázar and Torralba 2015).

The results show that all Scorpion variants outperform the top three IPC 2018 planners in per-domain comparisons. We also see that Scorpion benefits from using SYS-SCP PDBs instead of the COMB heuristics in many domains. Using the union of both sets is clearly preferable to using either COMB or SYS-SCP alone, since it raises the total coverage to 1261 by 56 and 44 tasks, respectively. For maximum coverage (1265 tasks), Scorpion only needs SYS-SCP PDBs and Cartesian abstraction heuristics.

## 5 Conclusion

We introduced a new pattern selection algorithm based on saturated cost partitioning and showed that it outperforms

| | Complementary 1 | Complementary 2 | Delfi 1 | Scorpion | | | | Coverage |
|---|---|---|---|---|---|---|---|---|
| | | | | COMB | SYS-SCP | SYS-SCP+CART | SYS-SCP+COMB | |
| Complementary 1 | – | 7 | 4 | 12 | 9 | 9 | 9 | 1030 |
| Complementary 2 | **24** | – | 7 | 12 | 10 | 9 | 8 | 1093 |
| Delfi 1 | **35** | **28** | – | 16 | 15 | 13 | 13 | 1236 |
| COMB | **28** | **27** | **19** | – | 7 | 5 | 2 | 1205 |
| SYS-SCP | **29** | **25** | **21** | **15** | – | 4 | 4 | 1217 |
| SYS-SCP+CART | **29** | **26** | **22** | **16** | **10** | – | 4 | **1265** |
| SYS-SCP+COMB | **30** | **27** | **23** | **13** | **13** | **5** | – | 1261 |

Table 5: Comparison of IPC 2018 planners and Scorpion variants.

all other pattern selection algorithms from the literature. The algorithm selects a pattern if it is useful for *any* state in the state space. In future work, we would like to evaluate whether it is beneficial to restrict this criterion to a subset of states, such as all reachable states or a set of sample states.

## Acknowledgments

We thank the anonymous reviewers for their helpful comments. We have received funding for this work from the European Research Council (ERC) under the European Union's Horizon 2020 research and innovation programme (grant agreement no. 817639).

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

| Coverage | $h^{\text{SCP}}$ over different PDB heuristics | | | | | IPC 2018 Planners | | | Scorpion | | | |
|---|---|---|---|---|---|---|---|---|---|---|---|---|
| | HC | Sys-3-Lim | CPC | CEGAR | Sys-SCP | Comp1 | Comp2 | Delfi1 | Comb | Sys-SCP | Sys-SCP +Cart | Sys-SCP +Comb |
| agricola (20) | 0 | 0 | 0 | 2 | 0 | 6 | 5 | **13** | 2 | 3 | 3 | 4 |
| airport (50) | 36 | 23 | 33 | 36 | 38 | 21 | 24 | 27 | 39 | **46** | **46** | **46** |
| barman (34) | 4 | 4 | 11 | 4 | 11 | 11 | 11 | **16** | 4 | 11 | 11 | 11 |
| blocks (35) | 28 | 28 | 28 | 28 | 28 | 30 | 30 | **32** | 28 | 28 | 28 | 28 |
| childsnack (20) | 0 | 0 | 0 | 0 | 0 | 0 | 0 | **6** | 0 | 0 | 0 | 0 |
| data-network (20) | 12 | 11 | 13 | 13 | 13 | 12 | 13 | 13 | **14** | 13 | **14** | **14** |
| depot (22) | 11 | 12 | 11 | 13 | 13 | 7 | 7 | 11 | 14 | 14 | **15** | **15** |
| driverlog (20) | 13 | **15** | 13 | **15** | **15** | 13 | 14 | 14 | **15** | **15** | **15** | **15** |
| elevators (50) | 41 | 43 | 43 | 40 | 41 | 37 | 37 | **44** | 44 | 41 | **44** | **44** |
| floortile (40) | 6 | 11 | 15 | 11 | 16 | **34** | **34** | 29 | 16 | 33 | 32 | **34** |
| freecell (80) | 21 | 26 | 25 | 27 | 30 | 22 | 26 | 26 | **70** | 31 | 68 | 65 |
| ged (20) | **19** | **19** | **19** | **19** | **19** | 16 | **19** | 15 | **19** | **19** | **19** | **19** |
| grid (5) | **3** | **3** | **3** | **3** | **3** | 2 | 2 | **3** | **3** | **3** | **3** | **3** |
| gripper (20) | 8 | 8 | 7 | 8 | 8 | **20** | **20** | **20** | 8 | 8 | 8 | 8 |
| hiking (20) | 13 | 14 | **19** | 18 | 15 | 15 | 18 | 18 | 14 | 15 | 15 | 14 |
| logistics (63) | 30 | 33 | 30 | 34 | **36** | 26 | 28 | 28 | 35 | **36** | **36** | 35 |
| miconic (150) | 72 | 143 | 116 | 144 | 144 | 105 | 104 | 142 | **145** | **145** | 144 | **145** |
| movie (30) | **30** | **30** | **30** | **30** | **30** | **30** | **30** | **30** | **30** | **30** | **30** | **30** |
| mprime (35) | 24 | **34** | 23 | 31 | **34** | 19 | 21 | 23 | 30 | **34** | **34** | **34** |
| mystery (30) | 17 | **19** | 16 | **19** | **19** | 13 | 16 | 17 | **19** | **19** | **19** | **19** |
| nomystery (20) | **20** | **20** | **20** | **20** | **20** | 12 | 19 | 18 | **20** | **20** | **20** | **20** |
| openstacks (100) | 49 | 53 | 49 | 49 | 53 | 74 | 73 | **89** | 51 | 53 | 53 | 53 |
| organic (20) | **7** | **7** | **7** | **7** | **7** | **7** | **7** | **7** | **7** | **7** | **7** | **7** |
| organic-split (20) | 10 | 10 | 10 | 10 | 10 | 13 | 13 | **14** | 13 | 13 | 13 | 13 |
| parcprinter (50) | 40 | 32 | 34 | 32 | 41 | 38 | 41 | 48 | **50** | **50** | **50** | **50** |
| parking (40) | **13** | 8 | 9 | 12 | **13** | 2 | 2 | **13** | **13** | **13** | **13** | **13** |
| pathways (30) | 4 | 4 | 4 | **5** | **5** | **5** | 4 | **5** | **5** | **5** | **5** | **5** |
| pegsol (50) | **50** | 48 | 48 | 48 | 48 | 48 | 48 | 48 | **50** | 48 | 48 | **50** |
| petri-net (20) | 0 | 7 | 4 | 4 | 8 | 16 | 18 | **20** | 0 | 7 | 6 | 0 |
| pipes-nt (50) | 21 | 24 | **25** | 23 | 23 | 15 | 22 | **25** | **25** | 23 | **25** | **25** |
| pipes-t (50) | 18 | 18 | 17 | 17 | 18 | 13 | 16 | **22** | 18 | 18 | 18 | 18 |
| psr-small (50) | **50** | **50** | **50** | **50** | **50** | **50** | **50** | **50** | **50** | **50** | **50** | **50** |
| rovers (40) | 8 | 8 | 10 | 9 | 10 | **14** | 13 | **14** | 11 | 12 | 13 | 13 |
| satellite (36) | 6 | 6 | 7 | 8 | 9 | 11 | 10 | **14** | 9 | 10 | 10 | 10 |
| scanalyzer (50) | 23 | 37 | 25 | 31 | **41** | 21 | 21 | 34 | 33 | **41** | **41** | **41** |
| snake (20) | **14** | 12 | 13 | 12 | 13 | 11 | 13 | 11 | 13 | 13 | 13 | 13 |
| sokoban (50) | **50** | **50** | **50** | **50** | **50** | 46 | 48 | **50** | **50** | **50** | **50** | **50** |
| spider (20) | 15 | 14 | 14 | 14 | 14 | 10 | 11 | 11 | **16** | 14 | 15 | **16** |
| storage (30) | 16 | 16 | 16 | 16 | 16 | 15 | 15 | **19** | 16 | 16 | 16 | 16 |
| termes (20) | 12 | 12 | 13 | 12 | 13 | 14 | **15** | 10 | 13 | 13 | 12 | 12 |
| tetris (17) | 10 | 11 | 11 | 10 | 11 | 10 | **13** | **13** | **13** | **13** | **13** | **13** |
| tidybot (40) | 22 | 28 | 24 | 26 | 30 | 24 | 29 | 29 | **32** | 30 | **32** | **32** |
| tpp (30) | 6 | 7 | 12 | 12 | 12 | 9 | **14** | 10 | 8 | 12 | 12 | 12 |
| transport (70) | 35 | 34 | 36 | 34 | 36 | 27 | 29 | 31 | 35 | 36 | **37** | **37** |
| trucks (30) | 9 | 9 | 10 | 13 | 12 | 12 | 10 | 12 | 12 | 13 | **16** | **16** |
| visitall (40) | 28 | **30** | **30** | **30** | **30** | 16 | 21 | **30** | **30** | **30** | **30** | **30** |
| woodwork (50) | 29 | 44 | 39 | 36 | 49 | 45 | 46 | **50** | **50** | **50** | **50** | **50** |
| zenotravel (20) | **13** | **13** | **13** | **13** | **13** | **13** | **13** | 12 | **13** | **13** | **13** | **13** |
| **Sum (1827)** | 966 | 1088 | 1055 | 1098 | 1168 | 1030 | 1093 | 1236 | 1205 | 1217 | **1265** | 1261 |

Table 6: Number of tasks solved by different planners.

Katz, M., and Domshlak, C. 2008. Optimal additive composition of abstraction-based admissible heuristics. In Rintanen, J.; Nebel, B.; Beck, J. C.; and Hansen, E., eds., *Proceedings of the Eighteenth International Conference on Automated Planning and Scheduling (ICAPS 2008)*, 174–181. AAAI Press.

Katz, M., and Domshlak, C. 2010. Optimal admissible composition of abstraction heuristics. *Artificial Intelligence* 174(12–13):767–798.

Korf, R. E., and Felner, A. 2002. Disjoint pattern database heuristics. *Artificial Intelligence* 134(1–2):9–22.

Korf, R. E.; Reid, M.; and Edelkamp, S. 2001. Time complexity of iterative-deepening A$^*$. *Artificial Intelligence* 129:199–218.

Korf, R. E. 1997. Finding optimal solutions to Rubik's Cube using pattern databases. In *Proceedings of the Fourteenth National Conference on Artificial Intelligence (AAAI 1997)*, 700–705. AAAI Press.

Lelis, L. H. S.; Stern, R.; and Sturtevant, N. R. 2014. Estimating search tree size with duplicate detection. In Edelkamp, S., and Barták, R., eds., *Proceedings of the Seventh Annual Symposium on Combinatorial Search (SoCS 2014)*, 114–122. AAAI Press.

Pearl, J. 1984. *Heuristics: Intelligent Search Strategies for Computer Problem Solving*. Addison-Wesley.

Pommerening, F., and Seipp, J. 2016. Fast Downward dead-end pattern database. In Muise, C., and Lipovetzky, N., eds., *Unsolvability International Planning Competition: planner abstracts*, 2.

Pommerening, F.; Helmert, M.; Röger, G.; and Seipp, J. 2015. From non-negative to general operator cost partitioning. In *Proceedings of the Twenty-Ninth AAAI Conference on Artificial Intelligence (AAAI 2015)*, 3335–3341. AAAI Press.

Pommerening, F.; Röger, G.; and Helmert, M. 2013. Getting the most out of pattern databases for classical planning. In Rossi, F., ed., *Proceedings of the 23rd International Joint Conference on Artificial Intelligence (IJCAI 2013)*, 2357–2364. AAAI Press.

Pommerening, F. 2017. *New Perspectives on Cost Partitioning for Optimal Classical Planning*. Ph.D. Dissertation, University of Basel.

Rovner, A.; Sievers, S.; and Helmert, M. 2019. Counterexample-guided abstraction refinement for pattern selection in optimal classical planning. In Lipovetzky, N.; Onaindia, E.; and Smith, D. E., eds., *Proceedings of the Twenty-Ninth International Conference on Automated Planning and Scheduling (ICAPS 2019)*. AAAI Press.

Seipp, J., and Helmert, M. 2014. Diverse and additive Cartesian abstraction heuristics. In Chien, S.; Fern, A.; Ruml, W.; and Do, M., eds., *Proceedings of the Twenty-Fourth International Conference on Automated Planning and Scheduling (ICAPS 2014)*, 289–297. AAAI Press.

Seipp, J., and Helmert, M. 2018. Counterexample-guided Cartesian abstraction refinement for classical planning. *Journal of Artificial Intelligence Research* 62:535–577.

Seipp, J., and Helmert, M. 2019. Subset-saturated cost partitioning for optimal classical planning. In Lipovetzky, N.; Onaindia, E.; and Smith, D. E., eds., *Proceedings of the Twenty-Ninth International Conference on Automated Planning and Scheduling (ICAPS 2019)*. AAAI Press.

Seipp, J.; Pommerening, F.; Sievers, S.; and Helmert, M. 2017. Downward Lab. https://doi.org/10.5281/zenodo.790461.

Seipp, J.; Keller, T.; and Helmert, M. 2017a. A comparison of cost partitioning algorithms for optimal classical planning. In Barbulescu, L.; Frank, J.; Mausam; and Smith, S. F., eds., *Proceedings of the Twenty-Seventh International Conference on Automated Planning and Scheduling (ICAPS 2017)*, 259–268. AAAI Press.

Seipp, J.; Keller, T.; and Helmert, M. 2017b. Narrowing the gap between saturated and optimal cost partitioning for classical planning. In *Proceedings of the Thirty-First AAAI Conference on Artificial Intelligence (AAAI 2017)*, 3651–3657. AAAI Press.

Seipp, J. 2017. Better orders for saturated cost partitioning in optimal classical planning. In Fukunaga, A., and Kishimoto, A., eds., *Proceedings of the 10th Annual Symposium on Combinatorial Search (SoCS 2017)*, 149–153. AAAI Press.

Seipp, J. 2018. Fast Downward Scorpion. In *Ninth International Planning Competition (IPC-9): planner abstracts*, 77–79.

Sievers, S.; Katz, M.; Sohrabi, S.; Samulowitz, H.; and Ferber, P. 2019. Deep learning for cost-optimal planning: Task-dependent planner selection. In *Proceedings of the Thirty-Third AAAI Conference on Artificial Intelligence (AAAI 2019)*. AAAI Press.

Torralba, Á.; Linares López, C.; and Borrajo, D. 2018. Symbolic perimeter abstraction heuristics for cost-optimal planning. *Artificial Intelligence* 259:1–31.

Yang, F.; Culberson, J.; Holte, R.; Zahavi, U.; and Felner, A. 2008. A general theory of additive state space abstractions. *Journal of Artificial Intelligence Research* 32:631–662.
