# OpenReview forum: "Pattern Selection for Optimal Classical Planning with Saturated Cost Partitioning"
_icaps-conference.org/ICAPS/2019/Workshop/HSDIP_

### Official Review · AnonReviewer2 · 2019-03-29
**somewhat incremental - well written - accept**

**Rating:** 7
**Confidence:** 4

**Review:**

The paper proposes a new method of selecting a "useful" subset of a systematically enumerated set of patterns (PDBs), subject to their use with saturated cost partitioning.

The paper is perhaps somewhat incremental, but it has a new method and a solid experimental evaluation. I don't have too much to add.

Does the PatternUseful function attempt to rank how useful a new pattern is, in the context of an SCP sequence? That is, does it select the pattern that has the largest number or fraction of abstract states with non-zero cost, or that have the largest increase in the total estimate, or anything like this, or simply the first pattern that makes any improvement at all? (This is like the difference between a steepest ascent and a greedy hill-climbing search.)

One question is regarding the comparison between the Scorpion planner and the other IPC-2018 planners. From the results in Table 5, even the baseline Scorpion planner, which, unless I'm mistaken, is the same that participated in the 2018 IPC, outperforms the other three, by a margin of about 10% greater coverage. However, in the IPC results, the result is the converse, with both Complementary versions and Planning-PDBs both having about 10% greater coverage than Scorpion. Is this only an effect of the domain selection? That is, that the set of domains and/or instances used in the competition is different from (a strict subset of?) the set used in this experiment. This could be clarified.

In general, the presentation format, which combines number of domains with higher/lower coverage and the total coverage is a good compromise to summarize a coverage comparison, but it does leave out the magnitude of differences in individual domains.

There are a few broken section references. References to section numbers do not work in the AAAI format, since sections are not numbered. There is one on page 4, section "Sys-SCP Pattern Selection Algorithm": "(see Section )". Another is in the subsection "Comparison of Pattern Selection Algorithms": "...from Section :".

---

### Official Review · AnonReviewer1 · 2019-04-05
**Nice paper**

**Rating:** 8
**Confidence:** 4

**Review:**

The paper proposes a new method to the automatic generation of pattern
collections for PDB heuristics. The method incrementally combines a set
of patterns using a saturated cost-partitioning and only adds a
pattern to the final selection if it increases the heuristic value
returned for any state, or is able to detect dead-end states that are
not detected by any other pattern in the collection. The authors prove
that these two conditions can be checked only based on the current
pattern collection without having to argue about all concrete states
in the state space, which makes their approach applicable in practice.
In the experimental evaluation, the new pattern selection is combined
with existing methods, leading to a substantial increase in coverage.

I found the paper to be very well written, the proofs seem correct,
and the experimental results are quite impressive.

My main two comments are the following:

1) I don't see conceptually why it would be a good idea, when checking
if a pattern is useful, to only use the current pattern sequence
\sigma to see if the heuristic improved. In particular in the
beginning of an execution of the inner loop, when \sigma does not
yield a good heuristic yet, this will add any pattern to C, no matter
if it improves the collection C or not. If I got it right, it might
even be that a pattern P is added to C that is subsumed by another
pattern P' \in C, is this correct? Thus, I don't see why adding such
patterns to C is necessarily good, since there is no guarantee that
the heuristics resulting from C is improved. Wouldn't at least a
subsumption check against patterns in C make sense? I got that this
facilitates the "PatternUseful" check.

2) I don't agree with your hypothesis that many small patterns are
always beneficial over few large patterns. While I agree that this is
good in many domains, there are clearly others where you need to have
large patterns to capture essential interaction between the variables
of a task. It is also somewhat contradicted by your experimental
results, where "states-down" is slightly better than "states-up", so
prefering large abstract states spaces seems to better. Does it make
sense to try to combine your focus on small patterns with a pattern
ordering that introduces *some* large patterns into the collection?


Minor things:
- reference to numbered sections (I guess from IJCAI?) are broken.
- in Def.8, you use the notation T(\tau) for the transitions which is
not defined.
- in the proof of Theorem 1, you talk about the "preimage" without
introducing it.

---

### Author Response · Authors · 2019-04-08
**Thank you for your constructive comments!**

Review 1:

> 1) I don't see conceptually why it would be a good idea, when checking
> if a pattern is useful, to only use the current pattern sequence
> \sigma to see if the heuristic improved. In particular in the
> beginning of an execution of the inner loop, when \sigma does not
> yield a good heuristic yet, this will add any pattern to C, no matter
> if it improves the collection C or not. If I got it right, it might
> even be that a pattern P is added to C that is subsumed by another
> pattern P' \in C, is this correct? Thus, I don't see why adding such
> patterns to C is necessarily good, since there is no guarantee that
> the heuristics resulting from C is improved. Wouldn't at least a
> subsumption check against patterns in C make sense? I got that this
> facilitates the "PatternUseful" check.

You are right, a pattern P might be added to C even if it is subsumed by
another pattern P' \in C. This is good thing, however, since subsumed
patterns can be useful in cost partitionings. If that wasn't the case,
there would not be any reason to keep patterns of size 1 when computing
a cost partitioning over systematic patterns of size 2. It is also
correct that additional checks could prove to be useful. In future work,
we would like to try whether it is beneficial to only add patterns that
improve the heuristic value for sample states. The Sys-SCP algorithm
presented in the paper has the advantage that it doesn't need to sample
states and instead will add a pattern if *any* state could benefit from
this. Consequently, it is less likely to ignore a helpful pattern.

> 2) I don't agree with your hypothesis that many small patterns are
> always beneficial over few large patterns. While I agree that this is
> good in many domains, there are clearly others where you need to have
> large patterns to capture essential interaction between the variables
> of a task. It is also somewhat contradicted by your experimental
> results, where "states-down" is slightly better than "states-up", so
> prefering large abstract states spaces seems to better. Does it make
> sense to try to combine your focus on small patterns with a pattern
> ordering that introduces *some* large patterns into the collection?

You are right, some tasks might need large patterns. We will tone down
our hypothesis. Considering large patterns in Sys-SCP or combining
Sys-SCP with a pattern selection algorithm that selects large patterns
could certainly be helpful.

> Minor things:
> - reference to numbered sections (I guess from IJCAI?) are broken.
> - in Def.8, you use the notation T(\tau) for the transitions which is
> not defined.
> - in the proof of Theorem 1, you talk about the "preimage" without
> introducing it.

Thanks for bringing this to our attention. This is fixed in the revised
version of the paper on OpenReview.



Review 2:

> Does the PatternUseful function attempt to rank how useful a new pattern
> is, in the context of an SCP sequence? That is, does it select the
> pattern that has the largest number or fraction of abstract states with
> non-zero cost, or that have the largest increase in the total estimate,
> or anything like this, or simply the first pattern that makes any
> improvement at all? (This is like the difference between a steepest
> ascent and a greedy hill-climbing search.)

The Sys-SCP algorithm adds the patterns greedily. Using a
steepest-ascent approach here is not feasible since there are usually
too many systematic patterns.

> One question is regarding the comparison between the Scorpion planner
> and the other IPC-2018 planners. From the results in Table 5, even the
> baseline Scorpion planner, which, unless I'm mistaken, is the same that
> participated in the 2018 IPC, outperforms the other three, by a margin
> of about 10% greater coverage. However, in the IPC results, the result
> is the converse, with both Complementary versions and Planning-PDBs both
> having about 10% greater coverage than Scorpion. Is this only an effect
> of the domain selection? That is, that the set of domains and/or
> instances used in the competition is different from (a strict subset
> of?) the set used in this experiment. This could be clarified.

Indeed, the difference stems from the selection of domains.
Scorpion-Comb is the planner that participated in IPC 2018. If we
consider only STRIPS IPC 2018 domains, the coverage scores are as
follows (see also the new table in the revised paper):

Complementary1: 86
Complementary2: 95
Planning-PDBs:  94
Scorpion-Comb:  76

> In general, the presentation format, which combines number of domains
> with higher/lower coverage and the total coverage is a good compromise
> to summarize a coverage comparison, but it does leave out the magnitude
> of differences in individual domains.

We completely agree. Since there was enough space we have added a table
holding per-domain coverage results for Tables 4 and 5 to the revised
version of the paper.

---

> ### Comment · AnonReviewer1 · 2019-04-10
> **Thank you for the response!**
>
> Thank you in particular for pointing out that subsumed pattern can actually be beneficial. I didn't think of that when reading the paper. I guess this deserves a short note in the final version.
>
> In the current approach, I really like that you can efficiently check if a pattern is useful for any state. Still, as you mention, it could be interesting to see if this can be complemented by checking the heuristic improvement on a set of sampled states.

---

### Meta-Review · Program_Chairs · 2019-04-25

**Recommendation:** Accept
**Confidence:** 5

**Metareview:**

Dear Authors,
thank you very much for your submission. We are happy to inform you that
we have decided to accept it and we look forward to your talk in the workshop.
Please, go over the feedback in the reviews and correct or update your papers
in time for the camera ready date (May 24).
Best regards
HSDIP organizers